# Influence of Spinal Shock on the Neurorehabilitation of ANNPE Dogs

**DOI:** 10.3390/ani12121557

**Published:** 2022-06-16

**Authors:** Débora Gouveia, Ana Cardoso, Carla Carvalho, Ana Rita Gonçalves, Óscar Gamboa, Rute Canejo-Teixeira, António Ferreira, Ângela Martins

**Affiliations:** 1Arrábida Animal Rehabilitation Center, Arrábida Veterinary Hospital, Azeitão, 2925-538 Setúbal, Portugal; anacardosocatarina@gmail.com (A.C.); mv.carla.c@gmail.com (C.C.); vetarrabida.lda@gmail.com (Â.M.); 2Superior School of Health, Protection and Animal Welfare, Polytechnic Institute of Lusophony, Campo Grande, 1950-396 Lisboa, Portugal; 3Faculty of Veterinary Medicine, Lusófona University, Campo Grande 376, 1749-024 Lisboa, Portugal; ana.rita.silva.goncalves@hotmail.com (A.R.G.); p5637@ulusofona.pt (R.C.-T.); 4Faculty of Veterinary Medicine, University of Lisbon, 1300-477 Lisboa, Portugal; ogamboa@fmv.ulisboa.pt (Ó.G.); aferreira@fmv.ulisboa.pt (A.F.); 5CIISA—Centro Interdisciplinar-Investigação em Saúde Animal, Faculdade de Medicina Veterinária, Av. Universidade Técnica de Lisboa, 1300-477 Lisboa, Portugal

**Keywords:** ANNPE, neurorehabilitation, dogs, spinal shock, nociception, spinal shock scale

## Abstract

**Simple Summary:**

Acute noncompressive nucleus pulposus extrusion (ANNPE) is characterized by a peracute onset of neurological clinical signs. It is based on the spinal cord contusion with possible manifestation of signs compatible with spinal shock. Treatment is conservative and may involve the implementation of physical rehabilitation protocols. Thus, this study intends to contribute to the knowledge surrounding the ANNPE research and to verify if the presence of spinal shock can affect the outcome, as well as recovery time after neurorehabilitation.

**Abstract:**

Acute noncompressive nucleus pulposus extrusion (ANNPE) is related to contusive spinal cord injuries, and dogs usually appear to be exercising vigorously at the time of onset. ANNPE has a characteristic peracute onset of clinical signs during exercise or following trauma, with non-progressive signs during the first 24 h and possibly signs of spinal shock. The main aim was to assess if the presence of spinal shock affects the neurorehabilitation outcomes of ANNPE dogs. This prospective controlled cohort clinical study was conducted at the Arrábida Rehabilitation Center. All of the dogs had T3–L3 injuries and were paraplegic/monoplegic with/without nociception, the study group (n = 14) included dogs with ANNPE spinal shock dogs, and the control group (n = 19) included ANNPE dogs without spinal shock. The study group was also evaluated using a new scale—the Spinal Shock Scale (SSS)—and both groups were under the same intensive neurorehabilitation protocol. Spinal shock was a negative factor for a successful outcome within less time. SSS scores > 4 required additional hospitalization days. The protocol was safe, tolerable, and feasible and accomplished 32% ambulation within 7 days, 29% in 14 days, and 29% in 30 days. The results were better than those obtained in previous studies—94% at 60 days—and 75% of the dogs without nociception recovered ambulation. Long-term follows-ups carried out 4 years later revealed a positive evolution.

## 1. Introduction

In the scientific literature, acute noncompressive nucleus pulposus extrusion (ANNPE) was previously known as high velocity–low volume disk extrusion, type III disk extrusion, traumatic intervertebral disk extrusion, or intervertebral disk explosion [1,2,3,4,5,6].

The ANNPE is predominantly related to contusive spinal cord injury (SCI). It usually presents in association with an external trauma, such as a car crash or a fall from a height [5], although reports have shown that at onset, most dogs are exercising vigorously. A tearing of the complex lamellar structure of the annulus fibrosus may occur along with a sudden material extrusion of the non-degenerated nucleus pulposus into the vertebral canal, suggesting that in dogs of increasing age, the annular lamellar is vulnerable [7].

Contusions may cause the release of local inflammatory mediators, cytotoxins, and excitatory neurotransmitters that promote a chain of secondary events and that lead to ischemia and the depletion of cellular energy [8,9]. In humans, it has been shown that elevated cytokine levels, such as the tumor necrosis factor alpha (TNF-α) and interleukin 1-beta (IL-1β) appear within minutes after injury and can last up to 2 h. Additionally, increased levels of glutamate, a neurotransmitter of the central nervous system (CNS), can be present, leading to astrocytic dysfunction [10].

Thus, ANNPE is characterized by a peracute onset of the non-progressive, often lateralized paresis/paralysis of one to four limbs depending on the neurolocalization of the injury [11]. The affected dogs often show vocalization at the onset of clinical signs and moderate spinal hyperesthesia noted on initial examination [12,13]. Additionally, the non-progression of clinical signs has been observed 24 h after injury [14,15]. This usually manifests as a highly asymmetrical T3–L3 myelopathy that affects medium and large dog breeds [9,16,17]. The prognosis for recovery in dogs with ANNPE that present with paraplegia without nociception is considered poor, and the majority of these dogs are euthanized within a week of injury [18].

The main differential diagnosis of ANNPE is the fibrocartilaginous embolic myelopathy (FEM), which has a similar clinical presentation [1,2,19]. MRI is the ideal imaging exam [3,16,19,20,21,22]; however, a definitive diagnosis is only possible through post-mortem histological examinations of the injured spinal cord segments [22,23]. The MRI findings can help to differentiate the ANNPE from FEM based on reductions in the volume and signal of the nucleus pulposus on T2-weighted (T2W) images, a focal hyperintense lesion within the spinal cord dorsally to the affected disc on T2W, an abnormal signal intensity in the epidural space, and a narrow intervertebral disc space with spinal cord compression that is absent or less than 10% [20,24].

On occasion, clinical signs may be considered compatible with spinal shock when there are temporary reduced spinal reflexes, including the withdrawal reflex in one or both hindlimbs. This is observed in the absence of any type of injury in the L4–S3 spinal cord segments or nerves during magnetic resonance imaging [16,18], which is a relevant clinical feature that may lead to an incorrect neuroanatomic localization [25].

Spinal shock syndrome has been described as a cause of the loss of motor and sensory function in both humans [26,27] and dogs [21,28,29] with severe SCI or in spinal cord transection. Spinal shock syndrome is a profound depression in the segmental spinal reflexes and a loss of muscle tone caudally to the lesion, even in physically intact reflex arcs (Jeffery and Smith 2005). It has also been described as involving the development of fecal and urinary incontinence [2,30,31].

Spinal shock syndrome is complex, and in dogs, the anal sphincter reflex typically recovers within 15 min of spinal cord transection; however, the withdrawal reflex may need from 2 days to 6 weeks to recover [32,33]. Furthermore, it manifests with a sudden interruption to the descending supraspinal input on motor neurons and interneurons as well as fusimotor inhibition and increased segmental inhibition [29]. For some authors, the end of spinal shock is considered to be the return of certain reflexes, and in humans, there is evidence of a slow transition from spinal shock to signs of spasticity [34].

ANNPE treatment is conservative and involves the implementation of physical rehabilitation [11,16,25,35] that does not require surgery [36] and that ideally includes an initial period of rest [17,37]. Some rehabilitation strategies may include kinesiotherapy exercises [37], electrical stimulation, and laser therapy [38,39,40,41]. Pharmacological management is based on opioids as well as NSAIDs such as paracetamol and gabapentin, which may be useful as additional analgesics in the early stages. Supportive care is essential, especially in the management of the urinary bladder, with possible need for manual expression [8].

Thus, the aim of this study was to contribute to the knowledge gaps in the field of ANNPE research and to verify if the presence of spinal shock syndrome may affect the neurorehabilitation outcomes of ANNPE dogs as well as time to recovery. It was hypothesized that this syndrome may delay the recovery of these patients. A second aim was to prove if intensive neurorehabilitation is safe, tolerable, and feasible in a clinical setting.

## 2. Materials and Methods

This prospective controlled cohort clinical study was conducted at the Arrábida Animal Rehabilitation Center (Setúbal, Portugal) between 1 February 2017 and 1 February 2022 after receiving approval from the Lusófona Veterinary Medicine Faculty (Lisbon, Portugal) ethics committee (No. 114-2021) and consent from the owners.

All of the dogs had to meet the following criteria for inclusion in the study with the onset of neurological dysfunction associated with vigorous exercise and T3–L3 lesions. All of the included dogs were paraplegic/monoplegic with or without nociception and therefore grade 1 or 0 according to the Modified Frankel Scale (MFS) [42], and characterized by peracute clinical signs, such as an asymmetric presentation, spinal discomfort, and evidence of spinal shock (reduced segmental spinal reflexes and muscle tone caudally to the injury) without L4–S3 lesions or peripheric nerve dysfunction. All of the dogs were diagnosed after computed tomography (CT), CT–myelography (Figure 1), or MRI and admitted to the rehabilitation center within the first 48 h after injury. Dogs were excluded if they presented with vertebral fractures, luxation, and/or compressive myelopathies. Additionally, the dogs diagnosed with presumptive FEM by a veterinary neurology specialist were immediately excluded from the study.

### 2.1. Study Design

In this present prospective controlled cohort clinical study, 33 dogs with thoracolumbar neuroanatomical localization received a neurorehabilitation consultation and were randomized by means of aleatory stratification according to the clinical signs that are consistent with spinal shock syndrome. They were divided into the study group (SG) (n = 14), which comprised ANNPE dogs manifesting signs of spinal shock, and a control group (CG) (n = 19) comprising ANNPE dogs without spinal shock. The total population characterization is described in Table 1 along with the registration of concomitant occurrences. The main occurrence was hip dysplasia associated with painful osteoarthritis of the coxofemoral joint, which was mainly observed in the SG (71.4%).

Dogs from both groups were subjected to an evaluation of their mental status, passive standing posture, spinal reflexes (patellar reflex, cranial tibial reflex, withdrawal reflex, crossed-extensor reflex, and perineal reflex), cutaneous trunci reflex, superficial pain, and deep pain perception (DPP). The DPP was evaluated in a controlled environment using 12 cm Halsted mosquito forceps in both the medial/lateral digits of both hindlimbs. This was also tested in the perineal region and at the tip and base of the tail. Spinal hyperesthesia was assessed with the vertebral palpation technique, which is performed by placing slight pressure on the vertebral column and one hand on the abdominal muscles, palpating throughout the ligamentum supraspinale between each spinous process (*processus spinosus*) and between each intervertebral space to activate the nociceptors located in the outer annulus fibrosus or of the dorsal longitudinal ligament.

An orthopedic examination including joint palpation distally to proximally was executed. Any signs of inflammation, edema, or crepitation and the presence of Ortolani in the hip that were observed were registered. Additionally, muscle tone was evaluated via the palpation of the quadriceps femoris, biceps femoris, semitendinosus and semimembranosus muscles, and the abdominal region (*rectus abdominis* and *obliquus externus abdominis*).

All of dogs were classified according to the Open Field Score (OFS) [43] at admission, and in the following time points, they were always in the same controlled 6 m floor, even if the dogs presented monoplegia without DPP. The SG was also evaluated using a pioneer scale that is currently in development to assess spinal shock—the Spinal Shock Scale (SSS) (Table 2).

The SSS is a functional punctuation scale that is based on the evaluation of three main spinal reflexes of the hindlimbs, the perineal reflex, the cutaneous trunci reflex, perineal tone, and hindlimbs tone. Thus, a dog with a classification score of 7 is considered to have severe signs of spinal shock.

Neurorehabilitation consultations and the following evaluations were always performed by a certified canine rehabilitation professional (CCRP) examiner/instructor. Data were recorded (Canon EOS Rebel T6 1300 D camera), and all videos were revised by two CCRP instructors.

The ambulatory status was considered according to OFS scores ≥ 11, although the dogs remained at the rehabilitation center for a maximum of 2 months to observe OFS improvements over time.

### 2.2. Procedures

Both groups were subjected to the same intensive neurorehabilitation protocol (INRP), which was based on electrical stimulation, laser therapy, and locomotor training.

#### 2.2.1. Electrical Stimulation

The electrical stimulation (BTL^®^ 4000 Smart, Famões, Portugal) programs included functional electrical stimulation (FES), a neuromodulation modality based on the application of superficial electrodes after trichotomy, according to the segmental technique [44,45,46]. All of the patients who showed an evident decrease in muscle tone were subjected to the co-contraction protocol, which consisted of the simultaneous contraction of both the quadriceps femoris group and the hamstring muscles group (Figure 2A). If the patients showed good muscle tone in the quadriceps femoris muscle tone associated with patellar hyperreflexia, then the electrodes were only placed following the sciatic nerve trajectory, stimulating the contraction of the hamstring muscles (Figure 2B). The current parameters were as follows: 40–60 Hz; 10–48 mA; 150 μs duty cycle 1:4; and trapezoid current [44,45]. Electrical stimulation was carried out for 10 min and was conducted first three times/day at first before decreasing to once a day (Table 3).

#### 2.2.2. Class IV Laser Therapy

Photostimulation programs using class IV laser therapy (Companion Therapy Laser—LiteCure^®^, New Castle, DE, USA) were performed with a 5 cm probe and transcutaneous application after trichotomy according to the neurolocalization of each dog. The protocol was the same as the one described by Bennaim et al. (2017) [47] and used 12 J/cm^2^ of energy, a wavelength of 810–987 nm, and the pulsed mode. The probe was applied to the injury site with slight pressure, and circulatory movements were always carried out in the cranial and caudal segments for one minute each (Figure 3). This protocol was performed daily for the first 15 days (Table 3).

Dogs with signs of hip osteoarthritis were treated with an 8–10 J/cm^2^ emission protocol and a four-point technique, which was also applied in continuous circulatory movements and 2 to 3 times a week [48,49]. Figure 4 depicts all of the points: the first point was located at the proximo cranial region of the hip joint; the second point was located at the proximo caudal; the third point was located at the disto caudal; and the fourth point was located at the disto cranial region.

#### 2.2.3. Kinesiotherapy Protocol

Locomotor training

Locomotor exercise was initiated on the second day after admission, and all of the dogs performed land treadmill (Superior Fit Fur Life Treadmill^®^, Fernhurst, Surrey, UK) and underwater treadmill (Hidro Pysio®, Broseley, UK) exercises (Figure 5). However, in some dogs with poor adaptation, bipedal training was implemented until the dogs were able to complete the exercises. Because all of the dogs were paraplegic/monoplegic, in the few first days of exercise, three rehabilitators helped to maintain postural standing and performed bicycle movements with the plegic limb(s) on the treadmill with the help of a body-weighted harness device.

The first sessions were performed on the land treadmill over a short period of time and several times a day (i.e., 3–5 min, 6–8 times/day) and gradually increased to longer training, i.e., 40 min, 2–3 times/day. The speed was also adjustable, from 1.8 km to 4.5 km/h, and the inclination was 2–10%. Land treadmill exercise was performed 6 times/week, and the underwater treadmill exercise was performed 5 times/week, according to the specifications described in Table 3.

Kinesiotherapy exercises

On the second day after admission, the kinesiotherapy exercise circuit, the one described in Table 3, was initiated. First, passive, and assisted exercises were carried out, always with the help of a rehabilitator.

All of the dogs performed a set of obstacle rails (5–6 times/day) via the stimulation of the monoplegic limb by the dorsal region of the paw touching the obstacle (Figure 6A). Additionally, gait stimulation using alternating floors, namely sand, grass, gravel, and soft ground (1–2 min; 3–6 times/day) (Figure 6B), and postural standing on a balance board (1–2 min; 3–6 times/day) (Figure 7A) were also implemented. These were followed by assisted bicycling while standing on a central stimulation pad with a rough surface (15 bicycle movements, 2–3 times/day) (Figure 7B). All of the exercises were carried out 5 days a week and, according to the frequency of the exercises, were decreased until they were only being performed 2–3 times a day, which was determined according to the dogs’ neurological evolution.

#### 2.2.4. Supportive Care

Hydric support was maintained for each dog at a rate of 100–150 mL/kg, and nutritional care was supported with an increased intake of 30%, according to the needs of each patient. Manual expression of the bladder was completed 3–4 times a day, and urine was monitored with regard to color, smell, and quantity. All of the dogs performed passive range of motion (PROM) exercises, completing 15–30 repetitions per session three times a day (Figure 8A), followed by massage (stroking, effleurage, wringing-up, kneading, friction, wringing-up, effleurage, and stroking) for a total of 5 min two times a day (Figure 8B).

### 2.3. Outcomes and Follow-Ups

The SG and CG dogs were assessed by neurological examination, as described in the flow cohort diagram (Figure 9). Assessments were always carried out by the same certified CCRP examiner/instructor. The measured outcomes for the SG were the OFS and the SSS scales, and for the CG, the only measured outcome was the OFS. The time points of evaluation were admission (T0), day 1 (T1), day 2 (T2), day 3(T3), day 5 (T4), day 7 (T5), day 14 (T6), day 21 (T7), day 30 (T8), and day 60 (T9).

Follow-ups were performed after clinical discharge at 7 days (F1), 15 days (F2), one month (F3), three months (F4), six months (F5), one year (F6), two years (F7), three years (F8), and four years (F9).

### 2.4. Data Collection

For all of the dogs (n = 33), data collection included the following categorical binominal variables: breed, sex, type of complementary exams carried out, neuroanatomical localization, presence of lateralization signs, spinal discomfort, nociception, fecal and urinary incontinence, occurrences, and flexion/extension pattern. The continuous quantitative variables included age, weight, number of days to achieve ambulation (categorized in ≤14 days, >14 to ≤30 days and >30 days), number of days in hospital (categorized as ≤14 days, 30 days, and 60 days), and number of days to recover DPP. Data collection also included categorical ordinal variables, such as OFS evaluation at admission, time points, and follow-ups, and ambulatory status with OFS scores ≥ 11. Ambulation was defined as the patient’s ability to stand up, maintain postural standing, take at least ten steps without assistance or weight support on any walking surface, and obtain voluntary or automatic micturition and defecation [44].

The SSS evaluation of the study group was carried out at admission and at the remaining time points, with scores ≤4 being considered a possible better prognosis and >4 being considered a possible worst one.

The cut-off of 4 was established due to clinical evidence regarding spinal shock severity. Thus, until a score of 4 points, reduced hindlimb reflexes and muscle tone may be considered the standard for spinal shock, whereas a score of more than 4 points may be associated with difficulties and an increased time to achieve a positive outcome.

### 2.5. Statistical Analysis

Quantitative and categorical data were recorded in Microsoft Office Excel 365^®^ (Microsoft Corporation, Redmond, WA, USA) and processed using IBM SPSS Statistics 25^®^ (International Business Machines Corporation, Armonk, NY, USA) software. Continuous variables (age and weight) were examined for normal distribution using the Shapiro–Wilk test (n < 50). As the data were normally distributed, continuous variables were reported as the arithmetic means, minimum, maximum, standard deviation (SD), and standard error of mean (SEM) with a 95% confidence interval. Descriptive statistics were performed by comparing the clinical and outcome variables between dogs with ANNPE by considering the presence or absence of spinal shock and allowing study population characterization.

The two groups were compared using a one-way repeated measures variance analysis (ANOVA) that considered the “number of hospitalization days” and “number of days until ambulation”. Chi-square tests were also performed to identify the differences between the SG and the CG regarding “the presence of spinal shock” and “the number of hospitalization days”. Univariate analysis of variance was carried out by considering the OFS scores with time as a repeated measure.

## 3. Results

In this prospective controlled cohort clinical study, as mentioned above, of the 33 dogs with a presumptive diagnosis of ANNPE, 21.2% were mixed-breed dogs, and 78.8% were pure-breed dogs, including Labrador retrievers (n = 7), Portuguese water dogs (n = 3), bullmastiffs (n = 2), Yorkshire terriers (n = 2), Spitz (n = 2), golden retrievers (n = 1), dachshunds (n = 1), grand danois (n = 1), French mastiffs (n = 1), poodles (n = 1), whippets (n = 1), Jack Russell terriers (n = 1), Bouvier bernois (n = 1), German shepherds (n = 1), and pitbulls (n = 1).

Age and weight had a normal distribution (*p* = 0.088 and *p* = 0.127, respectively), according to the Shapiro–Wilk Normality Test (n < 50). In each group, data normality was also achieved with a mean age of 5.6 years SD (5.94) in the study group and 6.4 years (SD) 9.48 in the CG, whereas a mean weight of 26.9 kg SD (15.57) was achieved in the SG and a mean weight of 21.9 kg was achieved in the SD (14.38) in the CG. Table 4 reports the results of the descriptive analyses in both the SG and CG.

Ambulatory status was achieved in 94% (31/33) of the total population, with two dogs (**one from the SG**) being euthanatized due to signs of progressive myelomalacia, one after T4 (day 5) and one after T6 (day 14). In the total sample, of the six dogs that did not have DPP, 67% (4/6) recovered: three dogs in the first 3 days after INRP (**one from the SG; two from the CG**) and one dog after 30 days (**one from the CG**). The two dogs that did not recover from DPP achieved the flexion/extension locomotor pattern within ~7 days in the monoplegic limb, leading to spinal reflex locomotion (**two from the CG**).

Regarding time until ambulation of the total population, 32% (10/31) of the dogs became ambulatory within 7 days after starting the INRP (**one from the SG; nine from the CG**), 29% (9/31) became ambulatory within 14 days (**two from the SG; seven from the CG**), and the same number of dogs became ambulatory in 30 days (**eight from the SG; one from the CG**). Only three of the dogs remained non-ambulatory after 30 days (**one from the SG; two from the CG**).

In the study group, as for the relation between the presence of spinal shock syndrome and days until ambulation, there was an evident difference between groups, which was demonstrated by one-way ANOVA ((F (1, 29) = [6.109], *p* = 0.02). Furthermore, a similar difference was observed when comparing the SG and CG with regard to the number of hospitalization days and the presence of spinal shock syndrome (X2 (2, n = 33) = 8.806, *p* = 0.012).

The SSS scores of the SG dogs decreased over time (Figure 10 and Table 5). In addition, a strong significance was observed between higher SSS scores during the first 48 h and the number of hospitalization days (X^2^ (2, n = 14) = 7.000, *p* = 0.03).

According to the univariate analysis of variance, a significant difference was observed between the groups at each time point (*p* < 0.001; Adjusted R^2^ = 0.865), a finding that could be also verified in the OFS-estimated marginal means evolution chart (Figure 11).

The long-term follow ups, which were carried out for a maximum of 4 years, implied a general positive exponential evolution regarding the OFS-estimated marginal means over the follow-up period (Figure 12).

There were no registered adverse effects to the INRP, such as hemodynamic alterations, which were only observed in two dogs who showed signs of progressive myelomalacia.

## 4. Discussion

In the present study, variability was observed in the breed distribution, with the Labrador retriever showing 21% prevalence (7/33), which is in agreement with Freeman (2020) [4]. This could be because it is a large non-chondrodystrophic breed, which is an important feature that has been referred to by McKee et al. (2010), Henke et al. (2013), and Mari et al. (2017) [22,35,37]. However, there was also an evident presence of small breeds, such as the Jack Russell terrier and Yorkshire terrier, as described by De Risio et al. (2009) [16]. The Portuguese water dog showed 9% (3/33) prevalence, as it is an autochthonous breed with a sample expression. Additionally, regarding sex, males were dominant in both groups, a finding that has been observed by other authors [2,16,22,24,50]. For the total sample population, the mean age was 6 years, and the mean weight was 24 kg, also similar to previous studies [11,16,22].

As for the SG and CG, the normality of the data for both age and weight were verified according to the Shapiro–Wilk Test, making the groups comparable.

Between groups, DPP recovery was mostly seen in the CG. For Dewey and Da Costa (2016) [51], ANNPE prognosis is favorable in dogs with DPP. Thus, from the eight dogs admitted to the study who did not have DPP, six of them became ambulatory (75%), with four recovering from DPP (50%) and two achieving spinal reflex locomotion (25%). These two dogs without DPP until clinical discharge showed signs of nociception recovery during the long-term follow-ups, a finding that is in contrast to many authors [16,18,24,37], and this can possibly be justified by the presence of intact trans-lesion connections [29,52,53]. Thus, ANNPE dogs without DPP at admission should be treated carefully according to their prognosis, but it may be possible to achieve a successful outcome. It is essential to identify the appropriate treatment for these dogs who show severe neurological signs [13].

The efficacy of INRP has already been proven in other situations, but the question remains as to how long dogs need to recover. Even with spinal shock syndrome, the total population of dogs revealed 32% (10/31) ambulation within the first 7 days of admission, 29% (9/31) ambulation in 14 days, and 29% ambulation in 30 days. As for the comparison between groups, according to the descriptive analysis, there is a clear delay in recovery in the SG. Previous studies showed successful outcomes of 67%, which were usually obtained within 1 month after injury [5,16], which can be compared to our results achieving 90 % (28/31) recovery within 30 days (T8) and 94% (31/33) recovery in 60 days (T9).

Thus, the differences between groups were evident, especially in regard to time until ambulatory status (*p* = 0.02), the number of hospitalization days (*p* = 0.012), and the OFS-estimated means at each time point registered by the univariate analysis of variance (*p* < 0.001; Adjusted *R*^2^ = 0.9) (Figure 1), demonstrating the negative relevance of spinal shock syndrome.

Considering the presence of spinal shock syndrome as a negative factor for a quick and successful clinical outcome, it is important for the neuro-rehabilitator to rely on a scale that is able to indicate the possible prognosis and time needed for recovery, i.e., the SSS. This evaluation is mainly essential in the 48 h after injury because within this timeframe, the first signs appear and may shift, with the possible quick recovery of the anal sphincter reflex after 15 min but with the withdrawal reflex only recovering any time from 2 days to 6 weeks [31,32,54]. In our study, only two dogs were euthanized after 5 and 14 days. These dogs had fecal and urinary incontinence at admission, signs that may be present as reported by other authors [50].

Therefore, the INRP may have a role in decreasing the time until recovery, and although the association between the presence of the withdraw reflex and the number of hospitalization days was not studied, the SSS score demonstrated a marked significance in relation to time (*p* = 0.03) in the SG, with dogs having an SSS > 4 requiring more days of hospitalization. On the CG, there was no need to apply this scale, as the two groups were separated after randomized stratification in regard to the presence of spinal shock.

Additionally, spinal shock areflexia may be present for a variable length of time and is followed by a gradual return of reflexes, possibly shifting to a state of hyperreflexia [29,55]. This state can be treated by resorting to electrical stimulation by means of FES [56] to avoid secondary spasticity signs, something that has already been suggested in human patients as a result of the involvement of residual descending connections [57] and a strong contribution from the reticulospinal tract [58,59], possibly delaying recovery, which were signs that were not observed in our patients.

Thus, our results suggest that spinal shock syndrome may not interfere in achieving ambulatory status, which is supported by other studies [18,22,31], although it may delay the time to complete recovery and require more hospitalization days, affecting the neurorehabilitation outcome and agreeing with our hypothesis.

Different authors have recommended restricted activity and cage rest for 4–6 weeks after injury [11,13,37,60], which is the opposite of our study, although the implementation of our INRP was carried out gradually regarding the type, frequency, and intensity of the exercises. This study has established a safe, tolerable, and a feasible protocol that can be applied in ANNPE dogs in a clinical setting. No side effects were observed, except in two dogs, who were determined to be showing myelomalacia progression.

The long-term follow-ups, which were carried out for maximum of 4 years, also revealed the positive evolution of the OFS-estimated marginal means in both groups, again with slightly lower values in the SG (Figure 12), which is indicative of the importance of spinal shock and SSS classification at admission, even for later recovery. The interpretation of the follow-up results should be conducted carefully due to the loss of the sample population after F6 (one year), decreasing the number of dogs who attended these consultations.

The limitations of this study included the reduced sample size, a lack of intra and inter-observers, no SSS validation, and the absence of CNS biomarkers. Additionally, it would be interesting to study the role of neurorehabilitation protocols in terms of hemodynamic alterations as well as fecal and urinary incontinence and their consequences when more dogs without DPP are included. Further studies should be continued with validation of the SSS.

## 5. Conclusions

This prospective controlled cohort clinical study showed a total of 94% recovery up to a maximum of 60 days, and these results are higher than those obtained in previous studies. Additionally, 75% of the dogs without DPP showed signs of recovery during the long-term follow-up period up to 4 years after intervention, revealing a positive evolution, results that can be accomplished in a clinical setting. However, the SG revealed a clinical and statistical delay compared to the CG regarding time until ambulation and the number of hospitalization days. Thus, it is suggested that the presence of spinal shock had a negative influence on the neurorehabilitation of ANNPE dogs.

## Figures and Tables

**Figure 1 animals-12-01557-f001:**
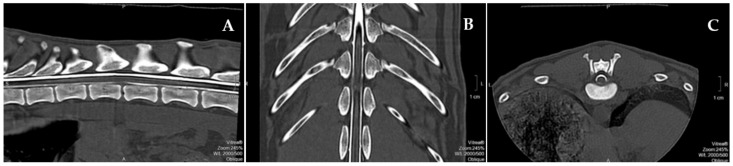
Computed Tomography–Myelogram image from a 6-year-old Labrador with T12–T13 acute non-compressive nucleus pulposus extrusion. (**A**) Vacuum phenomenon and light attenuation of the ventral contrast line diameter (sagittal view); (**B**) T12–T13 hyperintensity sign and bilateral light attenuation of the diameter of the contrast lines (dorsal view); (**C**) normal T12–T13 disk space with residual mineralized disk material ventrally (transverse view).

**Figure 2 animals-12-01557-f002:**
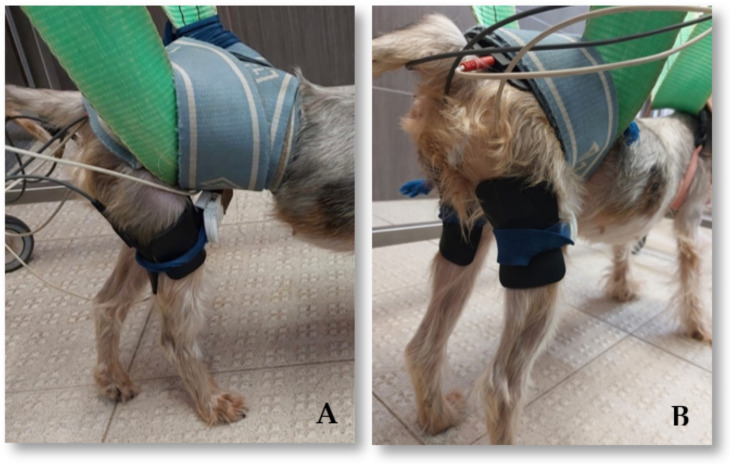
Electrical stimulation protocols on a dog in a postural standing position. (**A**) Co-contraction protocol with the stimulation of both the quadriceps femoris group and the hamstring muscles group. (**B**) Segmental technique of the sciatic nerve by means of functional electrical stimulation.

**Figure 3 animals-12-01557-f003:**
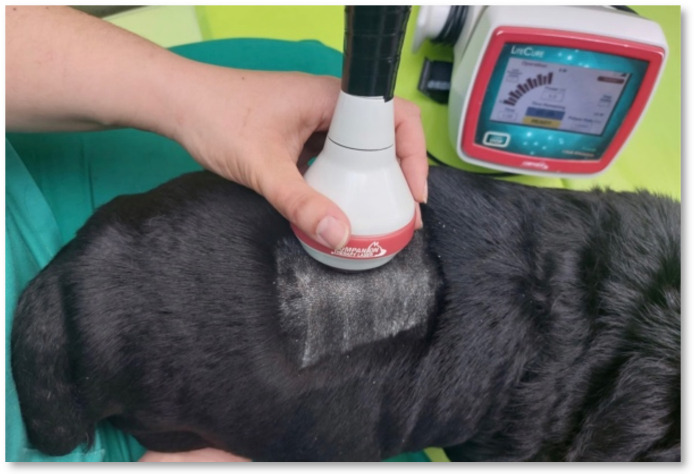
Laser therapy class IV program applied on the spinal cord injury.

**Figure 4 animals-12-01557-f004:**
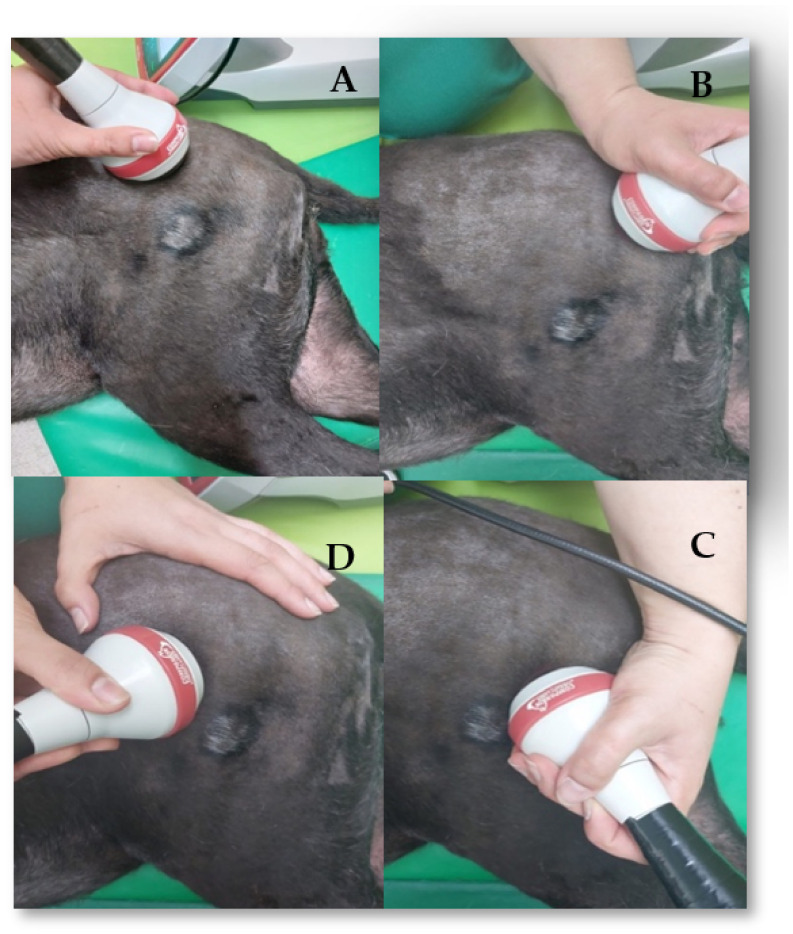
Laser therapy class IV program applied to the coxofemoral joint using the four-point technique. (**A**) Proximo cranial region; (**B**) proximo caudal region; (**C**) disto caudal region; (**D**) disto cranial region.

**Figure 5 animals-12-01557-f005:**
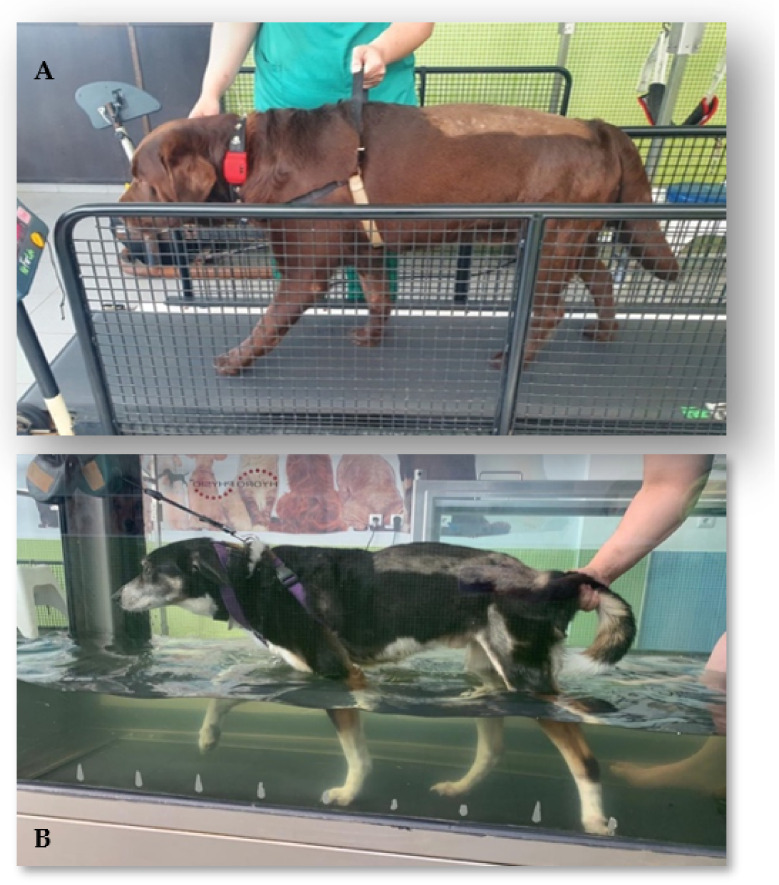
Locomotor exercise in two dogs. (**A**) Land treadmill; (**B**) underwater treadmill.

**Figure 6 animals-12-01557-f006:**
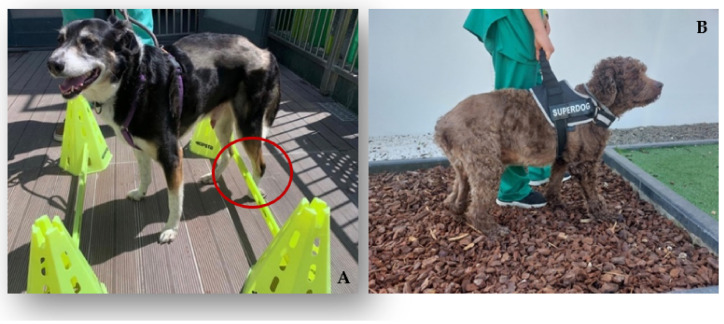
Kinesiotherapy exercises. (**A**) Obstacle rail; (**B**) gait stimulation on alternating floors; 
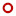
 knuckling position.

**Figure 7 animals-12-01557-f007:**
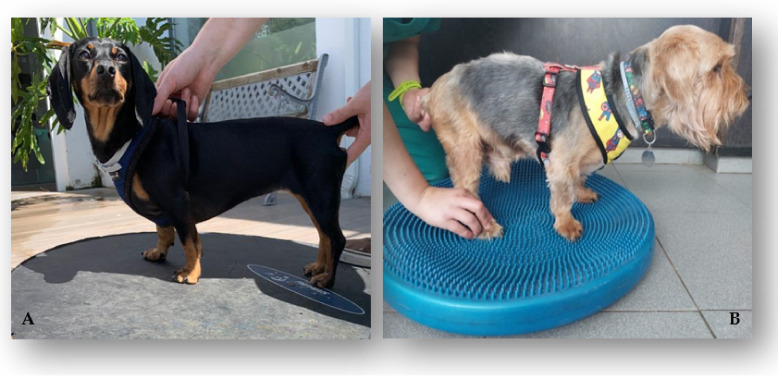
Kinesiotherapy exercises. (**A**) Postural standing on a balance board; (**B**) bicycle movements on a central stimulation pad.

**Figure 8 animals-12-01557-f008:**
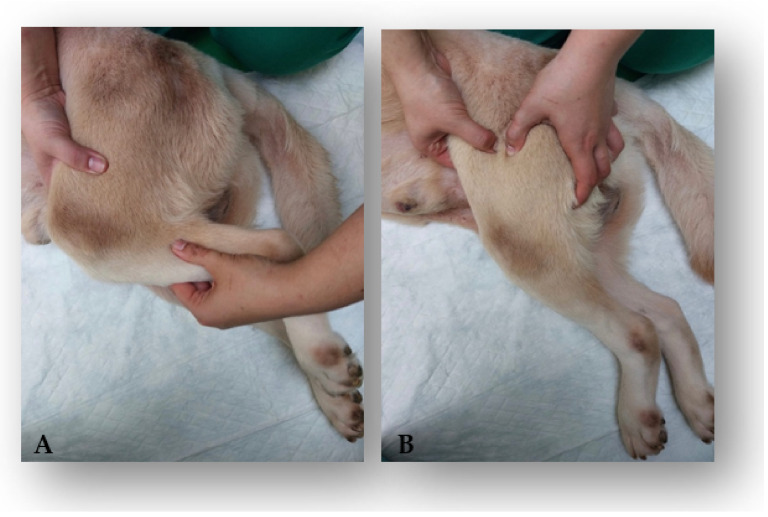
Supportive treatment. (**A**) Passive range of motion exercises for the knee joint; (**B**) deep kneading massage technique for direct muscular treatment.

**Figure 9 animals-12-01557-f009:**
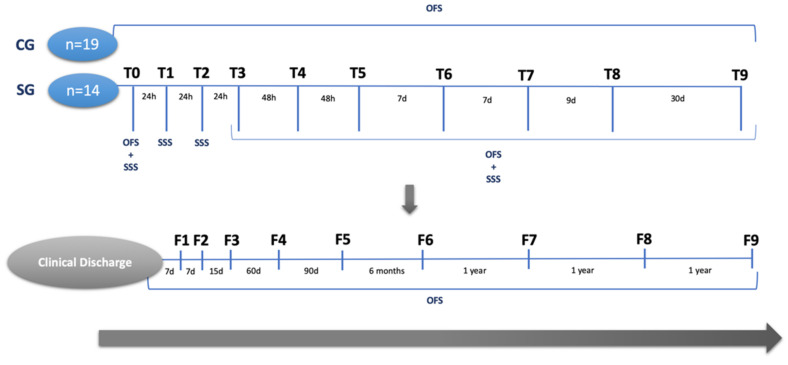
Flow cohort diagram describing the outcome measures of both the control and study group throughout the rehabilitation period and after clinical discharge. OFS: Open Field Score; SSS: Spinal Shock Scale; CG: control group; SG: study group; T0 (day 0); T1 (day 1); T2 (day 2); T3 (day 3); T4 (day 5); T5 (day 7); T6 (day 14); T7 (day 21); T8 (day 30); T9 (day 60); F1 (7 days); F2 (15 days); F3 (one month); F4 (three months); F5 (six months); F6 (one year); F7 (two years); F8 (three years); F9 (four years).

**Figure 10 animals-12-01557-f010:**
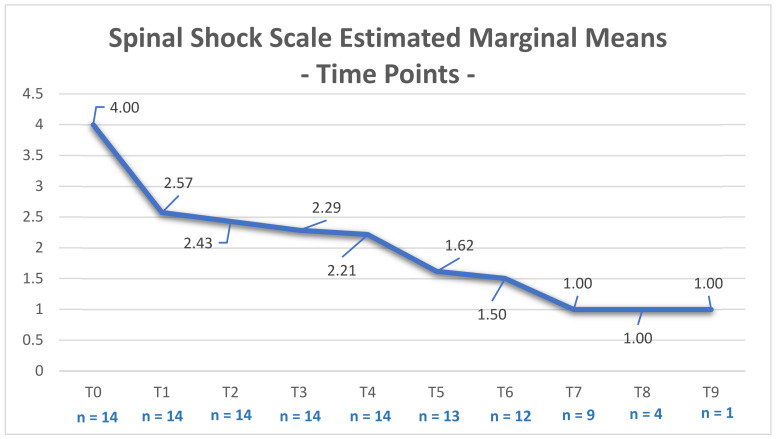
Evolution of the spinal shock scale-estimated marginal means in the study group throughout the intensive neurorehabilitation process. *Y*-axis: Spinal shock scale score; T0 (day 0); T1 (day 1); T2 (day 2); T3 (day 3); T4 (day 5); T5 (day 7); T6 (day 14); T7 (day 21); T8 (day 30); T9 (day 60).

**Figure 11 animals-12-01557-f011:**
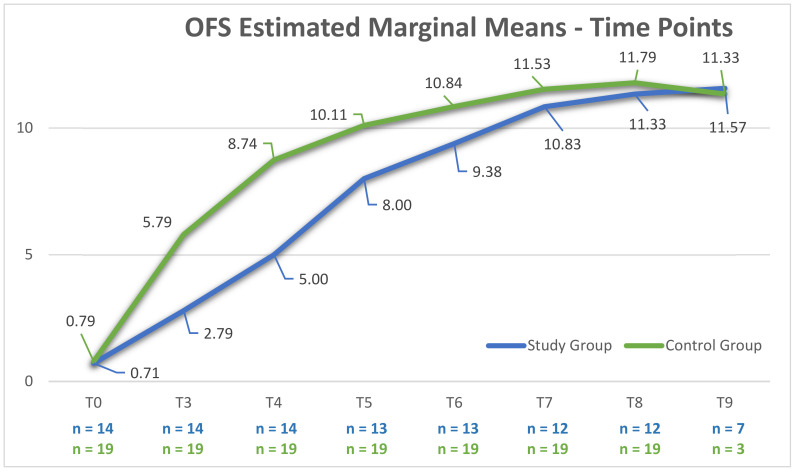
Evolution of the Open Field Score (OFS)-estimated marginal means in both the study and control groups throughout intensive neurorehabilitation. *Y*-axis: OFS; T0 (day 0); T1 (day 1); T2 (day 2); T3 (day 3); T4 (day 5); T5 (day 7); T6 (day 14); T7 (day 21); T8 (day 30); T9 (day 60).

**Figure 12 animals-12-01557-f012:**
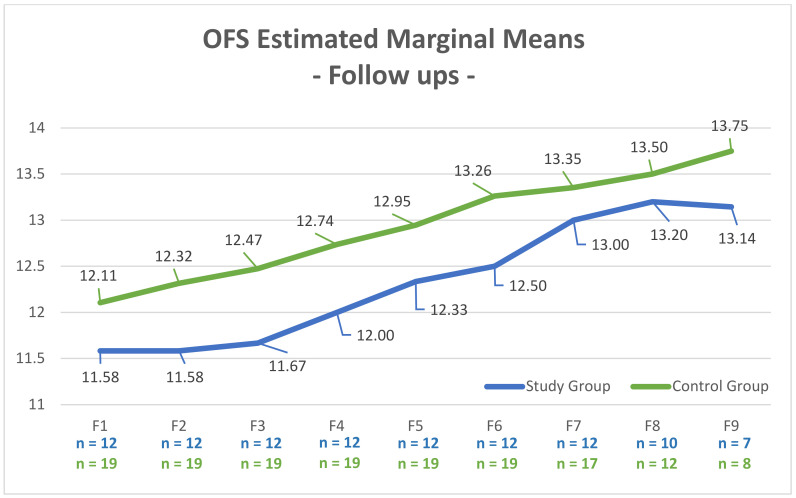
Evolution of the OFS-estimated marginal means in both the study and control groups during the follow-up periods. *Y*-axis: OFS (Open Field Score); F1 (7 days); F2 (15 days), F3 (one month), F4 (three months), F5 (six months), F6 (one year), F7 (two years), F8 (three years); F9 (four years).

**Table 1 animals-12-01557-t001:** Sample characterization of both the control and study groups.

	SG (n = 14)	CG (n = 19)	Total (n = 33)
Breed	Breed: 64.3% (9/14)	Breed: 89.5% (17/19)	Breed: 78.8% (26/33)
	Mixed Breed: 35.7% (5/14)	Mixed Breed: 10.5% (2/19)	Mixed Breed: 21.2% (7/33)
Sex	Male: 50% (7/14)	Male: 57.9% (11/19)	Male: 54.5% (18/33)
	Female: 50% (7/14)	Female: 42.1% (8/19)	Female: 45.5% (15/33)
Age	<7 years old: 71.4% (10/14)	<7 years old: 57.9% (11/19)	<7 years old: 63.6% (21/33)
	≥7 years old: 28.6% (4/14)	≥7 years old: 42.1% (8/19)	≥7 years old: 36.4% (12/33) Mean: 6 years
	Mean: 5.64 years	Mean: 6.42 years	
Weight	≤15 kg: 35.7% (5/14)	≤15 kg: 36.8% (7/19)	≤15 kg: 36.4% (12/33)
	>15 kg: 64.3% (9/14)	>15 kg: 63.2% (12/19)	>15 kg: 63.6% (21/33)
	Mean: 26.86 kg	Mean: 21.89 kg	Mean: 24 kg.
Complementary Exams	MRI: 35.7% (5/14)	MRI: 42.1% (8/19)	MRI: 39.4% (13/33)
	CT: 64.3% (9/14)	CT: 57.9% (11/19)	CT: 60.6% (20/33)
Neuroanatomical Localization	T12–T13: 14.3% (2/14)	T12–T13: 21.1% (4/19)	T12–T13: 18.2% (6/33)
	T13–L1: 35.7% (5/14)	T13–L1: 36.8% (7/19)	T13–L1: 36.4% (12/33)
	L1–L2: 21.4% (3/14)	L1–L2: 31.6% (6/19)	L1–L2: 27.3% (9/33)
	L2–L3: 28.6% (4/14)	L2–L3: 10.5% (2/19)	L2–L3: 18.2% (6/33)
Lateralization signs	Absent: 14.3% (2/14)	Absent: 21.1% (4/19)	Absent: 18.2% (6/33)
	Present: 85.7% (12/14)	Present: 78.9% (15/19)	Present: 81.8% (27/33)
Spinal discomfort	Absent: 78.6% (11/14)	Absent: 84.2% (16/19)	Absent: 81.8% (27/33)
	Present: 21.4% (3/14)	Present: 15.8% (3/19)	Present: 18.2% (6/33)
Nociception	Absent: 28.6% (4/14)	Absent: 21.1% (4/19)	Absent: 24.2% (8/33)
	Present: 71.4% (10/14)	Present: 78.9% (15/19)	Present: 75.8% (25/33)
Modified Frankel Scale	Grade 0: 28.6% (4/14)	Grade 0: 21.1% (4/19)	Grade 0: 24.2% (8/33)
	Grade 1: 71.4% (10/14)	Grade 1: 78.9% (15/19)	Grade 1: 75.8% (25/33)
Fecal and Urinary Incontinence	Present: 14% (2/14) (progressive myelomalacia)	Absent: 100% (19/19)	Present: 6% (2/33) (progressive myelomalacia)
	Absent: 86% (12/14)		Absent: 94% (31/33)
Occurrences	Absent: 28.6% (4/14) Present: 71.4% (10/14)	Absent: 57.9% (11/19) Present: 42.1% (8/19)	Absent: 45.5% (15/33) Present: 54.5% (18/33) (2 progressive myelomalacia)

Legend: SG (study group); CG (control group).

**Table 2 animals-12-01557-t002:** Spinal Shock Scale (SSS).

Reduced hindlimb reflexes	Withdrawal reflex	(+1)
	Patellar reflex	(+1)
	Cranial tibial reflex	(+1)
Reduced hindlimb tone		(+1)
Reduced perineal reflex		(+1)
Reduced perineal tone		(+1)
Absent/abnormal cutaneous trunci cut off		(+1)
TOTAL SCORE		

**Table 3 animals-12-01557-t003:** Intensive neurorehabilitation protocol.

KERRYPNX			Electrical Stimulation	Laser Therapy Class IV	Kinesiotherapy Exercises		
					Land Treadmill	Underwater Treadmill	Exercises
Admission to Ambulation	Study Group	DPP Positive	50 Hz, 10 mA, 150 μs, duty cycle 1:4;	After trichotomy, 12 J/cm^2^, 2,5 Hz, duty cycle 88%, 3 min, pulsed mode, SID, 15 days.	3–10 min;	5–10 min;	Obstacle rails, postural standing: 1–2 min, 3–6 times/day;
					no slope;	no slope;	bicycles: 2–3 times/day
					1.8 km/h	1.2–2 km/h	alternating floors: 1 min, 3–6 times/day,
			10 min; TID;		4–6 times/day;	SID;	6 days/week.
			SSS score < 4; BID.		6 days/week.	5 days/week.	
		DPP Negative	40 Hz; 10–48 mA; 150 μs,		3–10 min;	10–20 min;	Obstacle rails, postural standing: 1–2 min, 3–6 times/day;
						no slope;	
			duty cycle 1:4.		2–5% slope; 1.5 km/h	1.2–2 km/h	bicycles: 2–3 times/day;
			10 min; TID		6–8 times/day;	SID;	alternating floors: 1 min, 3–6 times/day,
			SSS score < 4; BID.		6 days/week.	5 days/week.	6 days/week
	Control Group	DPP Positive	50 Hz; 10 mA; 150 μs, duty cycle 1:4;		3–10 min;	5–10 min;	Obstacle rails, postural standing: 1–2 min, 3–6 times/day;
					no slope;	no slope;	bicycles: 2–3 times/day;
					1.8 km/h	1.2–2 km/h	alternating floors: 1 min, 3–6 times/day,
					4–6 times/day;	SID;	6 days/week.
			10 min; BID.		6 days/week.	5 days/week.	
		DPP Negative	40 Hz; 10–48 mA; 150 μs,		3–10 min;	10–20 min;	Obstacles rails, postural standing: 1–2 min, 3–6 times/day;
					2–5% slope; 1.5 km/h	no slope;	bicycles: 2–3 times/day;
					6–8 times/day;	1.2–2 km/h	alternating floors: 1 min, 3–6 times/day,
			duty cycle 1:4;		6 days/week.	SID;	6 days/week.
			10 min; BID.			5 days/week.	
Ambulation to Clinical Discharge	Study Group	DPP Positive	50 Hz; 10 mA; 150 μs, duty cycle 1:4;		10–40 min;	30 min;	Obstacle rails, postural standing: 1–2 min, 2–3 times/day;
					2–5% slope; 2–2.5 km/h	2–5% slope;	alternating floors: 2 min, 2–3 times/day,
					2–3 times/day;	2–2.5 km/h	3 days/week.
					3 days/week.	SID;	
			10 min; SID.			3 days/week.	
		DPP Negative	40 Hz; 10–48 mA; 150 μs,		10–40 min;	40 min;	Obstacle rails, postural standing: 1–2 min, 3 times/day;
			duty cycle 1:4;		2–5% slope; 1.8–2.5 km/h	5–10% slope;	alternating floors: 2 min, 3 times/day,
			10 min; SID.		2–3 times/day;	2.8–4.5 km/h	5 days/week.
					5 days/week.	SID;	
						5 days/week.	
	Control Group	DPP Positive	50 Hz; 10 mA; 150 μs, duty cycle 1:4;		10–40 min;		Obstacle rails, postural standing: 1–2 min, 2–3 times/day;
					2–5% slope; 2–2.5 km/h	30 min;	alternating floors: 2 min, 2–3 times/day,
					2–3 times/day;	2–5% slope;	3 days/week.
					3 days/week.	2–2.5 km/h	
						SID;	
			10 min; SID.			3 days/week.	
		DPP Negative	40 Hz; 10–48 mA; 150 μs,		10–40 min;	40 min;	Obstacle rails, postural standing: 1–2 min, 3 times/day;
					2–5% slope; 1.8–2.5 km/h	5–10% slope;	alternating floors: 2 min, 3 times/day,
					2–3 times/day;	2.8–4.5 km/h	5 days/week.
			duty cycle 1:4;		5 days/week.	SID;	
			10 min; SID.			5 days/week.	

Legend: DPP (deep pain perception); TID (three times a day); BID (twice a day); SID (once daily).

**Table 4 animals-12-01557-t004:** Descriptive analysis of age and weight in both the study and control group.

		**SG** **(n = 14)**	**CG** **(n = 19)**	**Total** **(n = 33)**
Age (years)	Mean	5.64	6.42	6.09
	Median	5	6	6
	Variance	5.94	9.48	7.898
	SD	2.437	3.079	2.81
	Minimum	2	1	1
	Maximum	11	12	12
	SEM	0.654	0.706	0.489
	Shapiro–Wilk Normality Test	0.55	0.326	0.088
Weight (kg)	Mean	26.86	21.89	24
	Median	24	21	21
	Variance	242.44	206.655	220.938
	SD	15.57	14.375	14.864
	Minimum	8	4	4
	Maximum	60	57	60
	SEM	4.161	3.298	2.587
	Shapiro–Wilk Normality Test	0.31	0.149	0.127

Legend: SG (study group); CG (control group); SD (standard deviation); SEM (standard error of mean).

**Table 5 animals-12-01557-t005:** Evolution of the SSS score in the study group over time.

	Time Points	T0	T1	T2	T3	T4	T5	T6	T7	T8	T9
ID											
**1**		6	2	2	1	1	1				
**2**		6	2	1	1	1	1	1	1		
**3**		3	2	2	2	1	1	1	1	1	
**4**		3	2	2	2	1	1	1	1		
**5**		3	2	2	2	2	1	1	1	1	
**6**		3	2	2	2	2	1	1	1		
**7**		3	2	2	2	2	2	1	1	1	
**8**		3	2	1	1	1	1	1			
**9**		3	2	2	2	2	1	1	1		
**10**		3	2	2	2	2	2	1	1	1	1
**11**		3	2	2	1	1	1	1	1		
**12**		3	2	1	1	1	1	1			
**13**		7	6	6	6	7	7	7			
**14**		7	6	7	7	7					

Legend: SSS (Spinal Shock Scale); ID (identification); T0 (day 0); T1 (day 1); T2 (day 2); T3 (day 3); T4 (day 5); T5 (day 14); T6 (day 14); T7 (day 21); T8 (day 30); T9 (day 60).

## Data Availability

The data presented in this study are available from the corresponding author upon request.

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
