# Peer review of "Influence of Spinal Shock on the Neurorehabilitation of ANNPE Dogs"

_animals, 2022, doi:10.3390/ani12121557_

Round 1

Reviewer 1 Report

General comments:

Intervertebral disc disease is the most common spinal cord disease of dogs. Over the years and primarily with the widespread use of CT and MRI, newer forms began to be recognized, such as acute noncompressive nucleus pulposus extrusion (ANNPE).

Using intensive neurorehabilitation intensive protocols the authors studied the influence of the spinal shock on the functional recovery following ANNPE in dogs.

Specific comments:

  1. Abstract, line 2: ANNPEs have a characteristic peracute onset of clinical signs during exercise or following trauma.
  2. Abstract: Delete the last sentence.
  3. Keywords: add “Spinal Shock Scale”.
  4. Introduction, page 1: please provide more information about the pathophysiology of ANNPE (tearing of the complex lamellar structure of the annulus fibrosus…).
  5. Introduction, lines 49-51: I do not understand the presence of this paragraph about ischemic myelopathy.
  6. Introduction, line 57: Please give more details about the MRI characteristics that can help to differentiate the ANNPE from an ischemic myelopathy.
  7. Materials and Methods, line 102: Did the authors also use CT-myelography?
  8. Materials and Methods, line 119: The perineal reflex is missing.
  9. Materials and Methods, line 136: The authors refer for the first time the Spinal Shock Scale (SSS). Is there any article published about it? It is crucial for the reader to know more about the SSS development and validation…
  10. Materials and Methods, line 186 and 199: Due to the fact that all dogs were paraplegic/monoplegic, it is important to describe the adaptations made by the authors to promote the adequate locomotor training and the kinesiotherapy exercises at the beginning of the rehabilitation.
  11. Results: It would be interesting to see some CT/Myelograms or MRI pictures of same clinical cases.

Reviewer 2 Report

The article needs major rewriting, but topic is important and worth reconsideration. I encourage the authors of the study to make the appropriate changes that follow.

Introducción

Description of ANNPE, although correct, is too large and not well organized. Please, follow a typical order (presentation, etiology, main sympthoms, etc) for the description and then, a short description of Spinal Shock syndrome. Two sypthoms have been mentioned compatible with spinal shock. However, other important sympthoms such as muscle tone caudal to the lesion and others must be cited (Smith and Jeffery, 2005).

Line 80, In my opinion a brief description of the most used protocols of physical rehabilitation for spinal disorders is necessary. Further down in the paragraph some pharmacological drugs have been mentioned.

Line 88, “to proof if intensive neurorehabilitation is safe, tolerable, and repeatable” have not been analyzed the study.

Although publications referred to the spinal cord disease and ANNPE has been shown, the current state of the research field and the gap of knowledge in what the study want to contribute have not been mentioned.

Line 89, “to verify…in less time”. This aim is implicit in the first one. Please, rewrite the Hypothesis and objective, they need to be clearly stated.

The Introduction section needs rewriting.

Materials and Methods

Line 93 (please, add province and country)

Line 99, ..MFS, pleas add Frankel et al. (1969) and add to the References

Line 112, add the main sympthoms of spinal shock to clarify

Line 126, According to the NAV (Nomina Anatomica Veterinaria) the ligamentum longitudinale dorsale connects the dorsal surfaces of the vertebral bodies, while the ligamentum supraspinale is attached to the tips of the spinous processes of the thoracic and lumbar vertebrae.

Line 127, periphery instead of peripheral.

Please, use latin terms for anatomical structures consistently throughout the text

Line 131, biceps femoris, semitendinous and semimembranous are not always flexor muscles. It depends on the phase of the stride. Just cite the muscles.

Line 134, OFS. Abbreviations should be defined the first time they appear in each of three sections: the abstract; the main text; the first figure or table. When defined for the first time, the abbreviation should be added in parentheses after the written-out form. Then OFS need to be described.

Line 152, add bibliography to the segmental technique or explain briefly.

Line 154, agonist and antagonist are referred to a specific action. However, some occasion they act like  synergist. Just mention the muscles.

Line 157, remove the term flexor and cite the muscles.

Line 162, remove the terms agonist and antagonist and cite the muscles.

Lines 177,178, 183 and 184, proximo instead of upper and disto instead of lower

Line 186, In my opinion the following paragraph describe what is also considered kinesiotherapy exercises. Dogs are not training, they are exercising and these exercises are part of the kinesiotherapy protocol. I recommend including among the kinesiotheraphy exercises.

Lines 189-193, It is not clear if this refer to land or underwater treadmill. Please rewrite to clarify.

Lines 191-193, rewrite in correct English.

Line 197, Please, remove quadrupedal locomotor training. I suggest “Land and underwater treadmill exercises.

Lines 227-228, The phrase is too subjective.

Data collection

The main issue of the study is that the statistical design does not correspond with the results and discussion sections. Besides, there is an inadequate description of the methods of statistical analysis. The objective of the chi-square test does not intend to evaluate the relations between variables. In the result and discussion sections “correlations” between variables have been mentioned and no statistical correlation study has been performed. What is more, the objective of the present study is not correlate variables, but identify differences in variables between groups. All this paragraph and result and discussion sections need rewriting. In case it is necessary the proper correlation analysis should be performed and result and discussion sections reorganized accordingly.

Results

Throughout the section paragraphs repeat what can be observed in tables and diagrams. Only the most outstanding results of the tables and diagrams should be mentioned in the text.

Line 289, Table 3 must be mentioned as showing the results down mentioned.

.Line 326, The diagram is not enough clear. Poorly written, needs major
rewriting for sense and flow

Lines 330-354, It is a succession of paragraphs without connection.

Line 330, “a tendency to significance”. This is not assumable in statistics

Line 332, no study of correlation has been mentioned in the statistic section

Discussion

As mentioned above one of my major concerns is the statistical design, then, the results, discussion and conclusions sections must be rewritten accordingly.

Conclusions: This section must be written in one or two paragraphs. In the present study is too large

Reviewer 3 Report

The authors describe a neurorehabilitation protocol for the treatment of ANNPE in dogs reporting overall good outcome in these patients. Evaluation of the influence of spinal shock on recovery was also considered in a subset of patients. Overall, information is delivered in an often non-linear stream of thoughts and with too much text that would benefit from English editing. Some tables for better comprehension of raw data, therapeutic protocols and clinical assessment would also improve the readability of the manuscript.

Lines 47-48: glutamate is a neurotransmitter of the CNS. It is not clear what the authors are stating with this sentence, please revise. Increased/decreased levels of glutamate?

Line 50: what does “vocal evidence of pain” mean? The term sounds misleading. Please revise the sentence. Also, in line 49 what is “This” referring to?

Lines 57-58: please replace “fibrocartilaginous embolism myelopathy” with “fibrocartilaginous embolic myelopathy”.

Line 60: please replace “spinal segments” with “spinal cord segments”.

Line 66: “transection” is a cross section along a long axis. Please replace with spinal cord transection.

Please provide a clear definition of spinal shock syndrome with an appropriate reference. Symptoms and causes are described here and there, but it is important for the reader to understand what is meant with this term.

Line 99: please provide a reference for the Modified Frankel Scale.

Line 101: please clearly state what is considered evidence of spinal shock.

Table 1: there are 2 patients in the study group presenting with progressive myelomalacia according to Table 1. The inclusion criteria states “all were…characterized by a peracute onset of non-progressive evolution of clinical signs”. Could the authors provide an explanation for this discrepancy?

Table 1: Modified Frankel score for the two groups should be reported.  

Line 119: please replace “peripheral reflexes” with “spinal reflexes”.

Line 127: please replace “peripheral fibrous annulus” with “outer annulus fibrosus”.

Line 134: please state what OFS stands for and provide a table of the parameters included for better comprehension.

The SSS is not validated, please state that too among the limitations in the discussion. It is not clear out of which calculation a cut-off of 4 was established. Also, was the SSS assessed by multiple clinicians? There is no clear overview on the scores of the animals of the study group either. Please consider a table reflecting the evolution of the score over time for the included animals.  

There is no clear overview on the intensive neurorehabilitation protocol (INRP). For some treatments (electrical stimulation) frequency and length of rehabilitation are not reported.  Summarizing the INRP in a table would be very useful for better comprehension. At which point and based on which criteria was the treatment discontinued or tapered off, or the animal discharged? Is there any specific definition for “ambulation” (assisted/non-assisted/+/-ataxia)?

Later in the results, it is reported that 2 dogs acquired involuntary spinal locomotion (spinal walking). Were these 2 animals included in the ambulatory group? Again see previous comment on definition of ambulation/recovery.

Figure 8: please adjust the writings of the Follow up timeline in English.

Lines 300-302: it cannot be distinguished when and how many animals from the 2 groups recovered the ambulatory status. Please clarify those results for the two groups.    

Figure 9: what was the SSS before T5?

Line 338: Also, the presence of dermatomes in T0…presented a strong significant relation with the number of hospitalization days. By definition: “A dermatome is an area of skin that is mainly supplied by afferent nerve fibres from the dorsal root of any given spinal nerve.” What is the message of this sentence? Do the authors mean sensation/cutaneous sensation? Same at Line 397.

The discussion is in general very long and information about the main findings is buried into too much text. The reader needs to go down to the conclusions in order to find clear statements about the findings of this study.   

Round 2

Reviewer 2 Report

The authors have revised thoroughly the article and substantial changes have been made that have improved the scientific quality of the study.  I encourage the authors to go on the researching line of this article and provide more similar studies

Author Response

We kindly appreciate the reviewer suggestion, however, and to our knowledge, there are no similar studies regarding neurorehabilitation and spinal shock, in association with this disease in particular. We have some studies only in regard to recovery that we have mentioned throughout the manuscript. Thank you so much for the possibility to enrich this manuscript, according to the reviewer suggestions. 

Reviewer 3 Report

The revised version of the manuscript is much improved in the overall organization and clarity. The authors adequately adjusted the manuscript and added tables that allow for clearer interpretation of their findings.

Author Response

We kindly appreciate the possibility to enrich this manuscript, according to the reviewer suggestions.